# A Synopsis Clonal Hematopoiesis of Indeterminate Potential in Hematology

**DOI:** 10.3390/cancers14153663

**Published:** 2022-07-28

**Authors:** Maroun Bou Zerdan, Lewis Nasr, Ludovic Saba, Paul Meouchy, Nadine Safi, Sabine Allam, Jenish Bhandari, Chakra P. Chaulagain

**Affiliations:** 1Department of Internal Medicine, SUNY Upstate Medical University, New York, NY 10009, USA; marounzerdan@gmail.com (M.B.Z.); bhandaje@upstate.edu (J.B.); 2Department of Hematology-Oncology, Myeloma and Amyloidosis Program, Maroone Cancer Center, Cleveland Clinic Florida, Weston, FL 33326, USA; 3Faculty of Medicine, Saint-Joseph University, Beirut P.O. Box 100, Lebanon; lewis.nasr@gmail.com (L.N.); ludovic.saba@gmail.com (L.S.); 4Department of Internal Medicine, Division of Hematology and Oncology, Naef K. Basile Cancer Institute, American University of Beirut Medical Center, Beirut P.O. Box 100, Lebanon; paulelmeouchy@gmail.com (P.M.); nadinesafi@gmail.com (N.S.); 5Department of Medicine and Medical Sciences, University of Balamand, Balamand P.O. Box 100, Lebanon; sabineallam@hotmail.com

**Keywords:** clonal hematopoeisis, chimeric antigen receptor T-cell therapy, mutations, hematology, myeloma

## Abstract

**Simple Summary:**

Mutations are not the norm, yet they exist. Having some mutations can infer information about a precancerous state. Clonal hematopoiesis of indeterminate potential is a condition of recurrent somatic mutations in the blood of otherwise healthy adults. In this review, we unravel the role of these mutations in multiple myeloma.

**Abstract:**

Clonal hematopoiesis of indeterminate potential can be defined as genetic mutations that correlate in hematologic neoplasia such as myelodysplastic syndrome. Patients with cytopenia increasingly undergo molecular genetic tests of peripheral blood or bone marrow for diagnostic purposes. Recently, a new entity has been demarcated to lessen the risk of incorrect diagnoses of hematologic malignancies. This new entity is a potential precursor of myeloid diseases, analogous to monoclonal gammopathy of undetermined significance as a potential precursor of multiple myeloma.

## 1. Introduction

Hematopoietic stem cells (HSCs) are replicating cells that are continuously exposed to multiple DNA-damaging agents that lead to the accumulation of somatic mutations [1]. Even though the majority of acquired mutations are phenotypically silent, some may provide a fitness advantage to replicating cells, resulting in selective proliferation defined as clonal expansion [2]. In individuals without established neoplasms, this process is defined as clonal hematopoiesis (CH). A subset of these proliferation-inducing somatic mutations occurs in genes that are associated with hematological malignancies such as leukemia, and have been termed CHIP [3]. Once present, there is a 0.5–1% per year risk of progression to a non-plasma cell hematologic neoplasm [4,5,6]. CHIP has also been associated with higher all-cause mortality mediated by increased risk of cardiovascular disease, myocardial infarction (MI), and stroke [3,7,8]. In this review, we highlight the growing role of CHIP in general, and then focus on hematologic malignancies, particularly multiple myeloma (MM).

## 2. Discussion

### 2.1. CHIP Detection Techniques and Variant Allele Frequency

While karyotyping, fluorescence in situ hybridization (FISH) and polymerase chain reactions (PCR) are all techniques used in CHIP detection; next-generation sequencing (NGS) is the mainstay for the diagnosis of CHIP due to its high sensitivity [9]. Whole exome sequencing (WES) or targeted gene panels are both used to search for somatic mutations in hematopoietic cells. Whole genome sequencing and WES are generally less sensitive as compared with targeted exome sequencing which looks for specific mutations in the coding region of genes known to be implicated in CHIP [10]. For example, it was noted that WES can detect clonal hematopoiesis (CH) in 10–15% of individuals older than 70 years (with variant allele frequency (VAF) of 3–10%) as compared with 25–75% prevalence when using targeted NGS [5,11,12]. However, NGS is characterized by a high error rate, reaching around 2% of the sequencing platform [13]. Therefore, using error-corrected targeted sequencing with greater coverage depth is able to identify clonal hematopoiesis (CH) harboring <1% VAF [11]. Thus, it can be inferred that the more precise and specific the detection technique is, the lower the number of VAF that can be determined.

An interesting observation in the past few years has been the detection of CHIP mutations in liquid biopsies analyzed for the management of solid tumors. Such mutations have also been found in bone marrow white blood cells, and therefore, their clonal origin has been confirmed. The presence of CHIP in liquid biopsies is particularly challenging and can cause confusion when detecting mutations specific for tumors due to hematopoietic cells infiltrating most tumors. However, CH mutations should not be simply discarded due to their differences from circulating tumor and free DNA, owing to the fact that several CH genes are known to carry oncogenic potential and can influence treatment options and response [12].

VAF is determined by NGS as the fraction of sequencing reads containing the mutation of interest out of the total number of reads from the same genomic location, and it usually differs according to the genes under study. The *DNMT3A*, *TET2*, and *ASXL1*, genes frequently encountered in CH, are usually detected with a VAF of 10–20%. On the contrary, *KRAS*, *GNAS*, *NRAS*, and *PIK3CA*, which are genes implicated in solid tumors, are often detected at considerably lower VAF values ranging from 0.1 to 0.5% [12]. While the cutoff used for VAF has been set at 2% for the diagnosis of CHIP, several studies have demonstrated varying effects on disease states with different VAFs and different genes. Using VAF > 10% correlated with the development of hematopoietic malignancy, but VAF < 10% did not show any increased risk [6,14]. Several studies have also demonstrated stability of the VAF of some genes over time versus expansion of other genes with clinical implication in the expanding genes. For instance, the *TET2* VAF increases with age, while the *DNMT3A* VAF generally remains stable [15,16].

### 2.2. Types of Clonal Hematopoiesis (CH) Mutations

The genes that are commonly mutated in CH are the same driver genes that are associated with acute myeloid leukemia (AML), myelodysplastic syndrome (MDS), and myeloproliferative neoplasms (polycythemia vera and myelofibrosis). Although their frequencies vary among studies, the three most common implicated genes are *DNMT3A*, *TET2*, and *ASXL1* (collectively known as “DTA mutations”) [6]. All three of these genes are epigenetic modulators, but two of them have opposing functions: while the *DNMT3A* enzyme catalyzes cytosine methylation, *TET2* catalyzes cytosine demethylation [10]. Other commonly mutated genes include *TP53*, *JAK2*, *SF3B1*, *GNAS*, *PPM1D*, and *BCORL1* [4]. These genes have a broad array of cellular functions including signal transduction, premRNA splicing, and DNA repair (see Table 1) [10,17].

A large population study showed that among 7216 individuals with CH, 69% had one mutation and 31% had two or more mutations. The spectra of CH mutations differed between oncogenes (which harbored missense mutations) and tumor suppressor genes (which exhibited truncating variants) [19].

The accumulation of somatic mutations throughout one person’s lifetime is the main driver towards CH [20]. In fact, mutations in the hematopoietic stem cell’s entire genome accumulate at a rate of 14 mutations per year, while mutations in the exome accumulate at a slower rate of 1 mutation per 10 years [9]. One study by Acuna-Hidalgo confirmed that there is an age-specific pattern of mutations in the driver genes. While C > T/G > A transitions were the most common overall, their incidence was not associated with age. Meanwhile, A > G/T > C mutations were most frequently found in adults over 45 years of age, accounting for 20% of all mutations in this subgroup. In contrast, younger individuals below 45 years had C > A/G > T as the most common transition, amounting to 30% of all their documented mutations [21].

CH without malignancy has also been reported in solid organs such as the esophagus, the colon, and the brain [22,23]. Regardless of the site, most mutations arise endogenously from (a) impaired DNA repair, (b) altered telomerase dynamics, or both [9]. Otherwise, these mutations can result from exogenous environmental mutagens such as smoking, air pollutants, radiation exposure, and chemotherapy [24]. Indeed, cancer therapy can predispose to the accumulation of mutations with clonal expansion. For instance, one study by Bolton et al. showed that mutations in ovarian and endometrial cancers had a significant skewing towards *PPM1D* mutations. However, when patients without a prior history of cancer therapy were excluded, the association with *PPM1D* decreased significantly and became similar to that of all other cancer types. This implies that both the underlying malignancy and the cancer treatment are at the interplay of acquiring somatic mutations that lead to CH [19].

### 2.3. CHIP’s Utility in Diseases

Between approximately 150,000 and 200,000 hematopoietic stem cells (HSCs) populate the human body, leading to the production of billions of cells every day. Over time, somatic mutations accumulate and can ultimately lead to CHIP [1]. CHIP describes apparently healthy individuals (without hematological malignancies) with somatic mutations at a variant allele frequency (VAF) greater than or equal to 2%. This cutoff may be modified in the future with major advances in genome sequencing capabilities [3].

Importantly, CHIP is an independent risk factor for myeloid malignancy, cardiovascular disease (CVD), and all-cause mortality. The association between CHIP and CVD has been under study since 2014 [6]. Regarding CVD, the hypothesis is that the myeloid cells, especially macrophages, that are derived from HSCs with CHIP-associated mutations release an excessive number of inflammatory cytokines. This causes chronic inflammation that provides these same cells with a functional and survival advantage. Inflammatory macrophages are instrumental to the pathogenesis of atherosclerosis. Therefore, the resulting positive feedback loop between CHIP and chronic inflammation increases the risk of development of atherosclerosis and chronic heart failure [18,25].

CHIP prevalence has also been studied in patients with autoimmune disorders, such as aplastic anemia, and states of chronic inflammation, such as human immunodeficiency virus (HIV). They have a statistically significant increase in the rates of CHIP as compared with healthy controls [26].

Aging is among the most prominent risk factors for CHIP. CHIP has been shown to be almost ubiquitous in those older than 85 years. The importance of age has held through genetic studies in elderly twins, showing no clear genetic predisposition or heritable variants of CHIP mutations [27]. There are other predisposing factors for CHIP such as history of chemotherapy or radiation, smoking, inflammation, and male sex [28]. The risk factors behind the etiology of CHIP can be memorized by a pneumonic CRIS (C = chemotherapy, R = radiation, I = inflammation and infection, S = smoking and male sex).

### 2.4. CHIP in Malignancies

#### 2.4.1. Chemoradiation

After aging, the second most important risk factor for CHIP is a history of chemotherapy and/or radiation. Some clonal selection pressures have been linked to genomic mutations. For instance, patients with platinum agent and doxorubicin exposure have higher rates of PPM1D mutations. Yet, this effect is not present after recovery from hematopoietic stem cell transplantation. On the one hand, *PPM1D* mutations have also been observed at higher rates within patients receiving radionuclide therapy, topoisomerase inhibitors, and taxanes. On the other hand, *CHEK2* mutations rates were only increased in platinum or topoisomerase inhibitors exposure [29].

Moreover, chemotherapy may be a method of progression of CHIP instead of an inductor of CHIP. This is postulated from the fact that most post-chemotherapy CHIP mutations can be detected prior to the administration of any treatment [30].

In a study assessing patients treated with chemotherapy, radiation, or immunotherapy, mutations remained stable in most patients despite their cancer treatments. In fact, mutated clones increased in 28% of patients. Interestingly, the presence of more than one mutation was associated with a statistically significant increase irrespective of mutation or treatment type [19]. Another study evaluated VAF evolution over time and found a difference in the way chemotherapy affected different genes. *DNMT3A* mutations remained stable while *TET2* mutations showed dynamic changes [31].

#### 2.4.2. Chimeric Antigen Receptor T-Cell Therapy (CAR-T)

Chimeric antigen receptor therapy (CAR-T) has become an eminent weapon in our treatment arsenal against numerous hematologic malignancies and its indications are still growing and evolving. In a study of 154 patients receiving CAR-T-cell therapy for hematological malignancies, CHIP was present in 48%. This could be at least partially attributed to that fact that these patients had had prior lines of chemotherapy and/or radiation therapy. Interestingly, the presence of CHIP in patients younger than 60 years was significantly associated with a higher likelihood of achieving complete response, even though there was no difference in progression-free survival or overall survival. The pathophysiology of this response warrants further investigation and warrants asking the question as to how specific CHIP mutations such as *TET2* and *DNMT3A* influence CAR-T function.

#### 2.4.3. Infection and Inflammation

Infection and inflammation have both been implicated in driving clonal expansion [32]. HIV is known to cause chronic inflammation and patients have a higher prevalence of CHIP. These higher rates of CHIP have also been observed in ulcerative colitis, rheumatoid arthritis, among other inflammatory conditions [33]. Conversely, patients with some CHIP variants seem to be predisposed to the development of infections [34]. More so, patients with CHIP who developed COVID-19 tended to have worse outcomes [35]. Therefore, the relationship between CHIP and inflammation certainly goes in both directions and may even form a positive feedback loop in certain situations. Yet, the relationship may be more complex than expected. For instance, a recently discovered inflammatory syndrome termed VEXAS (vacuoles, E1 enzyme, X-linked, auto-inflammatory, somatic), caused by somatic mutations in HSCs, does not seem to show the expected CHIP patterns, despite a hyperinflammatory state [36,37].

#### 2.4.4. CHIP in Therapy-Related Myeloid Neoplasms

Therapy-related myeloid neoplasms refer to those that occur in patients having already been treated with chemotherapy and/or radiotherapy. This includes treatment taken where the indication is a primary cancer or an autoimmune disorder. Improvements in gene sequencing techniques have allowed the proposal of the multi-hit model for the pathogenesis of therapy-related myeloid neoplasms. CHIP has been found to be a frequent first step leading to these secondary neoplasms. In fact, CHIP is often detectable prior to treatment initiation, therefore, possibly setting the fertile genomic ground for secondary leukemogenesis [37].

For instance, in one study, the presence of CH at the time of stem cell transplant in non-Hodgkin lymphoma patients resulted in a therapy-related myeloid neoplasm incidence of 14% vs. 4.3% in the patients without clonal hematopoiesis [38]. Moreover, it seems that the risk of developing secondary myeloid neoplasms increases with VAF and the number of mutations in the CHIP present [19]. Thus, there may be an indication to monitor patients receiving chemotherapy and radiation therapy for somatic mutations, and to require more frequent follow-up plan in those with CHIP mutations present.

Beyond the initial presence of CHIP, the evolution of cytotoxic treatment to secondary myeloid neoplasms is a complex affair. It is affected by treatment taken, the aging process itself, as well as individual exposures. The environment may add different hits, such as that relating to the TP53 gene or unfavorable karyotype abnormalities. For instance, the genetic mutation patterns observed in atomic bomb survivors seldom affect the expected sites of methylation and genes as compared with the usually studied populations. All in all, despite the continued development of the arsenal against secondary myeloid neoplasms, allogeneic stem cell transplantation remains the best currently available choice for curative intent in fit patients with unfavorable genetics [39].

#### 2.4.5. CHIP Associated with Adverse Outcomes

CHIP is known to be associated with increased all-cause mortality in both healthy patients and patients with cancer. For instance, CHIP has been associated with a statistically significant decrease in overall survival in patient with solid, non-hematological malignancies [24]. Moreover, it is a marker of poorer prognosis in patients with non-Hodgkin lymphoma as well as multiple myeloma (MM) receiving hematopoietic autologous stem cell transplantation (HSCT) [6]. In addition to its relationship with the onset of CVD previously discussed, CHIP has been associated with lower long-term survival and faster disease progression in patients with chronic heart failure [40].

### 2.5. CHIP in MM

When the diagnostic criteria of hematological malignancies are not met, but somatic hematopoietic mutations that are usually found in leukemia are present with a variant allele frequency (VAF) greater than 2%, we define this condition as CHIP [1]. This situation is particularly worth considering in MM for several reasons.

First, the existence of CHIP may represent a precursor state leading to the development of monoclonal gammopathy of unknown significance (MGUS) and subsequently to smoldering or asymptomatic MM, and then finally to symptomatic or active MM [2]. This was also documented in patients with Waldenström’s macroglobulinemia (WM) precursors; patients with CHIP progressed more to symptomatic WM [3].

In a study done by Peres et al., clinical, demographic, and molecular characteristics of patients with MM from different populations were explored. Patients were divided into three groups based on ethnicity: non-Hispanic blacks, non-Hispanic whites, and Hispanics. Twelve per cent of patients with MM in the cohort were found to have clonal hematopoiesis with VAF of 0.11. The presence of CH was statistically associated with an older age at diagnosis (65 vs. 60 years). CH was also significantly associated with higher ISS stage at diagnosis. The study team recommended further investigations regarding the association between CH and overall survival in non-Hispanic Black patient with MM and between CH and progression free survival in non-Hispanic White patients with MM [41].

Furthermore, the existence of CHIP results in poor clinical outcomes in MM patients. In fact, Mouhieddine et al. demonstrated that the myeloma progression rate of patients with CHIP after HSCT was higher than those without CHIP. This led to decreased overall survival (OS) and progression free survival (PFS). To interpret these results, several mechanisms may be at play. Patients with CHIP may have had increased toxicities related to the chemotherapy regimen, such as cytopenia, which could result in delayed treatment or infra-optimal dose usage. Another relevant factor could be the alteration of the microenvironment in the bone marrow due to proinflammatory cytokines, which could promote myeloma survival and progression. However, this finding was only observed in myeloma patients who did not receive long-term immunomodulatory drug (IMiD) lenalidomide maintenance after HSCT. The group who received lenalidomide maintenance had good outcomes regardless of CHIP presence or absence [4]. In contrast with MM, the presence of CHIP in WM was not correlated with a decrease in OS, and disease progression was not found to differ between patients with or without CHIP [3].

It is also worth mentioning that the presence of CHIP has also been shown to increase mortality by triggering an inflammatory reaction that promotes atherosclerosis and cardiovascular morbidity and mortality [5]. However, this was not demonstrated in patients with MM, in whom no increase in the incidence of cardiovascular mortality was observed [4]. This could be attributed to the intrinsic aggressiveness of MM [6]. Nevertheless, an increase in cardiovascular events was noted in patients with WM and CHIP [3].

The presence of CHIP has also beenassociated with worse outcomes in the presence of DNMT3A p.R882. However, the presence of CH did not seem to be correlated with the development of therapy-related myeloid neoplasms (TMN), which was more likely related to lenalidomide or thalidomide use in maintenance post-transplant. Data on this topic are still scarce and does not enable us to draw conclusions regarding the extent to which CHIP and associated VAF can contribute to TMN development, although mutations in TP53 were more prevalent in patients who developed TMN. DeStefano et al. postulated two ways in which CH could influence the development of MM. In CHIP, mutant hematopoietic stem cells can contribute to B cells that can acquire additional mutations leading to plasma cell malignancy. The presence of CH subsequently can influence disease phenotype and response to therapy. In addition, the presence of CHIP in mutant cells can alter the inflammatory cytokines and bone marrow microenvironment, thus, contributing to the progression of MM [42].

#### 2.5.1. Inflammation in CHIP

There is a strong association between CHIP and inflammation. CHIP is more prevalent among individuals with autoimmune and inflammatory conditions, such as anti-neutrophil cytoplasmic antibody (ANCA) vasculitis for example [8]. Studies have shown that the harmful effects of CHIP are in part related to inflammasome activation and endothelial injury [5]; both *TET2* and *DNMT3A* play an important role. In *TET2* -/- mice, there is an overproduction of the proinflammatory mediators such as interleukin (IL) 1 beta, IL6, and tumor necrosis factor (TNF) alpha, which are released from circulating cells. Similarly, the absence of the *DNMT3A* gene shifts the balance towards a proinflammatory state with an oversecretion of IL6, but also of C-X-C motif ligand (CXCL) 1, CXCL2, and C-C motif chemokine ligand 5 (CCL5). This applies to the peripheral circulation. At the marrow level, the absence of the *TET2* gene induces an expansion of hematopoietic stem cells and a remodeling of the microenvironment favorable to cell growth. Similarly, inflammation and fibrotic infiltration of the marrow are noted if the *DNMT3A* gene is absent [9]. Therefore, by stimulating the inflammatory process, CHIP alters the microenvironment in the bone marrow, which could explain the more severe progression in myeloma patients with CHIP [4]. However, this remains to be investigated and proven.

#### 2.5.2. Mutations and Clonal Selection

Genomic studies in multiple patients with various hematological malignancies have been conducted and have consistently found the involvement of epigenetic modifier genes. An important example is the DNA methyltransferase 3 alpha (*DNMT3A*) gene, which is frequently mutated in clonal hematopoiesis, and acts as a cytosine methylator. Another gene that has an opposing demethylating role is the tet methylcytosine dioxygenase 2 (*TET2*). These two genes are the site of more than 70% of all mutations identified in CHIP [6]. When methylation levels vary at gene promoters, transcription is affected and so is the preferential differentiation of cells [7]. In MM, mutations involving *DNMT3A* are also the most common, followed by *TET2*, *TP53*, *ASXL1*, and *PPM1D* [4]. Figure 1 summarizes the importance of these various genes during CHIP.

The different pathways are not mutually exclusive. When the *DNMT3A* and *TET2* genes are mutated, there is an epigenetic alteration of the DNA that leads to activation of the inflammasome. This proinflammatory state may play a role in the progression of MM [9]. The *PPM1D* gene mutation is mostly found in cell clones that have been exposed to chemotherapeutic agents [14]. *ASXL1* mutations are mostly associated with smoking [10].

As for the evolution of CHIP, the disease begins with a hematopoietic stem cell that acquires a somatic mutation. However, to achieve clonal expansion, it is necessary for this mutated cell to undergo a process of clonal selection which can be facilitated by various factors: chemotherapy, ionizing radiation, inflammation/infection, age, smoking, etc.

*ASXL1* mutations are mainly reported in smoking patients [10]. *ASXL1* is a gene that is frequently mutated during CH or hematological malignancies, even though it has no clear catalytic activity. It is thought to be involved in epigenetic regulation and histone scaffolding. Loss of function of this gene is associated with clonal proliferation [11]. It is thought that this proliferation is induced by the activation of the Akt/mTOR signaling pathway because the use of rapamycin, an inhibitor of this pathway, resulted in a decrease of this uncontrolled proliferation in mice [12].

Cancer treatments are not the source of the CHIP mutations. Research shows that the mutations were already present prior to the exposure to the cytotoxic agent. This suggests, instead, that chemotherapy promotes clonal expansion of mutated cells that have a survival advantage [10]. Cancer chemotherapy increases the frequency of mutations in the *PPM1D* and *TP53* genes. *PPM1D* is a gene that codes for the protein phosphatase 2C family serine-threonine phosphatase. Its role is largely to inhibit the tumor suppressor p53, and *PPM1D* expression is directly related to exposure to cytotoxic agents or ionizing radiation. Mutations in this gene are essentially categorized as gain of function [13]. Studies have shown that exposure to chemotherapeutic agents can enhance clonal selection of cells with mutations in *PPMD1* and *TP53* [14]. In fact, patients with refractory non-Hodgkin’s lymphoma (NHL) had a higher prevalence of *PPMD1* and *TP53* mutations than the otherwise healthy population [15,16]. Mouhieddine et al. investigated whether this was also the case in patients with MM and demonstrated that the prevalence of CHIP was lower in patients with MM as compared with those with refractory NHL. This was primarily attributed to the shorter duration of exposure to cytotoxic agents in MM patients but also the use of less toxic regimens [4]. Another important concept involves the therapy-related myeloid neoplasms (tMN). According to Gibson et al., the presence of CH before stem cell transplantation in patients with NHL increased the occurrence of tMN [17]. However, this was not the case for patients with MM [4].

## 3. Conclusions and Future Directions

Incorporating CHIP into clinical practice is still a work in progress. Nowadays, there are multidisciplinary teams that are counseling patients with CHIP on their risk of developing cardiovascular and hematological clinical outcomes. The first step in this approach should be offering a close follow-up of those patients. In addition, patients with CHIP should be advised on adjusting their modifiable risk factors (e.g., smoking cessation, optimization of body mass index, diabetes, and hypertension) and improving their lifestyle choices (e.g., healthy diet and exercise) [10].

Further studies on CHIP are needed due to the wide array of diseases associated with the condition, including but not limited to CVD, aplastic anemia, venous thromboembolic diseases, and hematologic malignancies. While CHIP also affects clinical outcomes in patients with solid tumors, mainly pertaining to the type of chemotherapy used and its mechanism in exerting selective pressure on gene mutations, understanding the interplay among CHIP, inflammation, and chemotherapy response will have implications on patient management in the upcoming years. Future studies should also focus on determining which CHIP patients would benefit from “prophylactic” treatment if any [18,43]. For example, *DNMT3A* R882 mutation, more than one mutation, or high VAF increase the risk of developing AML or CVD [44]. Other studies have showed vitamin C could mimic the effect of *TET2* [45]; mTOR inhibitors were also effective in eliminating clonal hematopoietic cells; while anti TGF-beta and anti-IL-6 could reduce inflammation associated with CHIP [46,47]. However, despite all these studies, it is not known, to date, who would benefit from preemptive treatment and who would not.

The clinical significance and possible causality of CHIP in MM is uncertain to date. Maia et al. used multidimensional flow cytometry to assess the presence of MDS-PA (MDS phenotypic alterations) and CH in newly diagnosed MM. Their findings demonstrated CH in 50% of MM patients with MDS-PA versus in one fifth of MM patients without MDS-PA; VAF was observed at 8%. The authors concluded that MDS-PA and CH were mostly present at diagnosis rather than an effect of chemotherapy. Larger cohorts assessing this relation are needed to better illustrate the role of CHIP in MM development and progression [48]. In addition, knowing morbidity carried by CHIP is important to assess in MM patients to better determine mortality causes and direct patient care in addressing conditions linked to CHIP.

Finally, it is important to differentiate among the different types of CHIP mutations if we want to associate them with clinical outcomes. For instance, a study by Ghobrial et al. retrospectively studied CHIP mutations in MM patients treated with HSCT. CHIP mutations in MM patients not receiving immunomodulator maintenance was associated with decreased PFS and OS. However, regardless of CHIP status, the use of immunomodulator maintenance was associated with improved PFS and OS [49]. The drawback of this study was that it looked at all CHIP mutations together and correlating them with clinical outcomes. Future studies should focus on studying each mutation alone as they may have different interactions with immunomodulators and clinical outcomes in patients with MM. This is also important clinically because having a particular CHIP mutation can dictate clinical decisions for MM patients, such as the decisions to give chemotherapy, immunomodulators, and even going for HSCT.

In terms of our case presented at the beginning of this paper, a few points are worth mentioning. The *PPM1D*, located on 17q23.2, encodes for a member of the PP2C family of serine/threonine protein phosphatases, which negatively regulate cellular responses to environmental stress, in part by inhibiting *TP53* activity. *PPM1D* mutations are more commonly found in patients with therapy-related myelodysplastic syndrome than primary myelodysplastic syndrome (15% versus 3%) and are associated with shorter survival after hematopoietic stem cell transplantation. The *PPM1D* mutations can also be found in clonal hematopoiesis of indeterminate potential (CHIP). In patients undergoing autologous stem cell transplantation for lymphoma, CHIP at the time of transplantation, particularly if *PPMD1* was mutated, has been reportedly associated with inferior survival and increased risk of therapy-related myeloid neoplasm (TMN) in the form of AML and MDS [39].

In a single institution, large retrospective study (N = 629), treated by HSCT after high dose melphalan conditioning, CHIP was present in 21.6% of cases with the most mutated genes being *DNMT3A*, *TET2*, *TP53*, *ASXL1*, and *PPM1D*. Unlike in the setting of relapsed NHL treated with HSCT, the presence of CHIP prior to HSCT for MM was not associated with an increased risk of TMN. Despite this, our patient was considered to be not a HSCT candidate, reflecting growing hesitancy from clinicians on offering HSCT to MM patients with concomitant CHIP. After the HSCT, a question may arise about what is the optimal maintenance strategy in MM patients with CHIP? In this study, ImiD maintenance improved PFS and OS for all patients regardless of CHIP status, however, IMiD maintenance therapy was significantly associated with developing a subsequent TMN in patients who had a CHIP. After a median of ~2.7 years of ImiD maintenance (lenalidomide or thalidomide), 3.3% of patients develop a TMN. Overall, the presence of CHIP, at the time of HSCT, does not increase the risk of TMN associated with IMiD maintenance, and patients with CHIP, when treated with IMiD maintenance, obtain a survival benefit similar to that seen in MM patients generally. Based on this, as of now, we should continue to offer HSCT and post-HSCT ImiD maintenance in MM patients with CHIP. As the survival of MM patients continues to improve, prospective long-term follow up data are required on impact of CHIP on survival and TMN, particularly in the current era of widespread use of anti-CD38 monoclonal antibody therapies (daratumumab and isatuximab) in the treatment of MM. Clinical trials of lenalidomide with or without daratumumab for post-HSCT maintenance therapy are ongoing (AURIGA-NCT03901963 and S1803-NCT04071457), which may shed light on the impact of daratumuamb on CHIP and subsequent development of TMN.

## Figures and Tables

**Figure 1 cancers-14-03663-f001:**
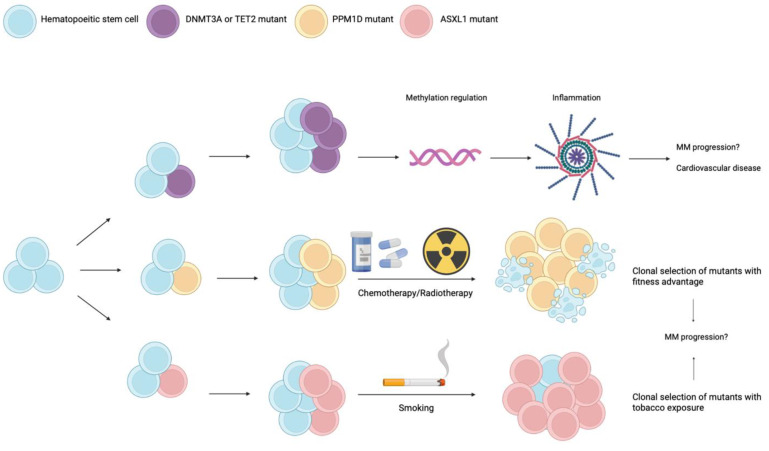
Schematic representation of the different mutations involved in CHIP.

**Table 1 cancers-14-03663-t001:** Functions of driver mutations involved in CH and other related conditions [10,17,18].

Gene	Function	Related Conditions
** *DNMT3A* **	Epigenetic regulator	UC, HIV, RA, solid tumors, ANCA-positive vasculitides, NHL or MM undergoing CAR-T, CVD, progression of CHF, AML, AA
** *TET2* **	Epigenetic regulator Inflammasome activation in *TET2* depletion	RA, solid tumors, ANCA-positive vasculitides, CVD, progression of CHF, AML
** *ASXL1* **	Epigenetic regulator	ANCA-positive vasculitides, CVD, AML, AA
** *JAK2* **	Signal transduction	CVD, PV, atherosclerosis aggravation
** *PPM1D* **	DNA damage response	Ovarian & Endometrial Cancers (specifically in treated patients); UC, NHL or MM undergoing CAR-T
** *TP53* **	DNA damage response	NHL or MM undergoing CAR-T, chemotherapy-induced clonal expansion
** *SF3B1* **	Spliceosome	MDS, bone marrow ringed sideroblasts, macrocytic anemia
** *SRSF2* **	Spliceosome	MDS

AA, AA Amyloidosis; AML, acute myeloid leukemia; ANCA, antineutrophil cytoplasmic antibodies; CAR-T, Chimeric antigen receptor therapy; CHF, congestive heart failure; CVD, cardiovascular disease; HIV, human immunodeficiency virus.

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
