# Peer review of "A Synopsis Clonal Hematopoiesis of Indeterminate Potential in Hematology"

_cancers, 2022, doi:10.3390/cancers14153663_

Round 1
Reviewer 1 Report
This is a fairly comprehensive review of CHIP, with specific reference to multiple myeloma. It is reasonably well written, although there is a degree of repetition in the document. I found it interesting and easy to read. There are a few other reviews on similar subject available, but this adds to the literature. I like the addition of the case review, especially as the outcome differs from the current evidence, which is discussed in the document.
One or two minor criticisms:
1. I dislike the simple summary, especially the opening sentence 'mutations are not the norm, but they exist', which means little as an opening introductory sentence. I would suggest re-writing this.
2. The introduction is sparse. Lines 118-129, later in the document, would make a better introduction if integrated (and aren't required for that segment of the review). Suggest integrate.
Author Response
- I dislike the simple summary, especially the opening sentence 'mutations are not the norm, but they exist', which means little as an opening introductory sentence. I would suggest re-writing this.
—> The required section has been rewritten.
2. The introduction is sparse. Lines 118-129, later in the document, would make a better introduction if integrated (and aren't required for that segment of the review). Suggest integrate.
—> The required section has been rewritten.

Reviewer 2 Report
This review by Zerdan, et al. reviewed the implications of clonal hematopoiesis of indeterminate potential (CHIP) in multiple myeloma. The authors introduced the detection methods for CHIP and types of CHIP mutations. They also described the association of CHIP and various diseases, including malignancy and inflammation, as well as the sources of CHIP. Lastly, a brief case with PPM1D E447 mutation was reported. The manuscript has two major problems that need to be modified.
1. The topic of this review is to discuss the role of CHIP in multiple myeloma. However, very little information is provided to suggest such relationship. Please provide more evidence to support that CHIP is important for predicting multiple myeloma.
2. The numbering and sectioning are unclear and thus lead to unclear structure. Please reorganize the Discussion part.
Author Response
Thanks a lot for your time, patience and consideration. We have addressed your comments. Please let us know if you have any further feedback!
- The topic of this review is to discuss the role of CHIP in multiple myeloma. However, very little information is provided to suggest such relationship. Please provide more evidence to support that CHIP is important for predicting multiple myeloma.
Thank you for your comment. We expanded the part in our discussion on CHIP and MM to include two more recent studies that discuss how CHIP can influence the development of MM.
- The numbering and sectioning are unclear and thus lead to unclear structure. Please reorganize the Discussion part.
The numbering and sectioning of the discussion are now modified to have a clearer flow and structure.

Round 2
Reviewer 2 Report
The manuscript has greatly improved with more details and clarifications. I recommend the paper for publication in the present form.
Author Response
Thanks a lot!
We highly appreciate your constructive feedback.
We will